# Chiral dinitrogen ligand enabled asymmetric Pd/norbornene cooperative catalysis toward the assembly of C−N axially chiral scaffolds

Liang Jin[1,2,3,4], Ya Li[1,4], Yihui Mao[1,4], Xiao-Bao He[2,3], Zhan Lu [1], Qi Zhang [1,2] ✉ & Bing-Feng Shi [1] ✉

C−N axially chiral compounds have recently attracted significant interest among synthetic chemistry community due to their widespread application in pharmaceuticals, advanced materials and organic synthesis. Although the emerging asymmetric Catellani reaction offers great opportunity for their modular and efficient preparation, the only operative chiral NBE strategy to date requires using half stoichiometric amount of chiral NBE and 2,6-disubstituted bromoarenes as electrophiles. We herein report an efficient assembly of C−N axially chiral scaffolds through a distinct chiral ligand strategy. The crucial chiral source, a biimidazoline (BiIM) chiral dinitrogen ligand, is used in relatively low loading and permits the use of less bulky bromoarenes. The method also features the use of feedstock plain NBE, high reactivity, good enantioselectivity, ease of operation and scale-up. Applications in the preparation of chiral optoelectronic material candidates featuring two C−N chiral axes and a chiral ligand for asymmetric C−H activation have also been demonstrated.

Atropoisomeric compounds are privileged scaffolds in pharmaceuticals, agrochemicals, bioactive natural products, and advanced materials[1–3]. They also serve as pivotal ligands and catalysts in organic chemistry[4,5]. In addition to the atropisomeric biaryls bearing C−C stereogenic axes, C−N axially chiral compounds have recently showcased their potential of being a novel type of axially chiral scaffolds that might have widespread application (Fig. 1A)[6–8]. However, catalytic asymmetric preparation of the latter has only gained limited success when compared with their C−C axially chiral siblings[9]. A fundamental challenge is the higher structural vulnerability of C−N chiral scaffolds originated from the increased degree of rotational freedom (Fig. 1B)[10–14], which has greatly limited the application of many synthetic tools. Ever since the seminal independent reports by Taguchi[15] and Curran[16], several strategies have been developed for the catalytic asymmetric preparation of such compounds, namely ring formation, C−N axes construction,

desymmetrization and rotational blocking of the C−N axes[11–29]. However, the utility of these elegant protocols has been inhibited by either substrate specificity or catalyst complexity. As a result, the development of novel and broadly applicable synthetic tools remains an urgent research topic.

The Catellani reaction, a type of palladium/norbornene (NBE) cooperative catalysis, has been recognized as a modular and highly efficient route toward highly functionalized molecules[30–36]. The reversible insertion of NBE has allowed the one-pot preparation of such molecules through functionalization of otherwise inaccessible *ortho*-C−H bond of iodoarenes with diverse elelctrophiles, followed by the *ipso*-coupling with various terminating reagents. Despite the powerfulness of this synthetic tool, controlling enantioselectivity in such a mechanistically complicated process hasn't met any success until very recently, by using chiral NBE (previously designed for functional group directed remote *meta*-C−H activation[37])[38–45], chiral

[1]Department of Chemistry, Zhejiang University, Hangzhou, China. [2]ZJU-Hangzhou Global Scientific and Technological Innovation Center, Zhejiang University, Hangzhou, China. [3]College of Chemical and Biological Engineering, Zhejiang University, Hangzhou, China. [4]These authors contributed equally: Liang Jin, Ya Li, Yihui Mao. ✉e-mail: qiqiyape@zju.edu.cn; bfshi@zju.edu.cn

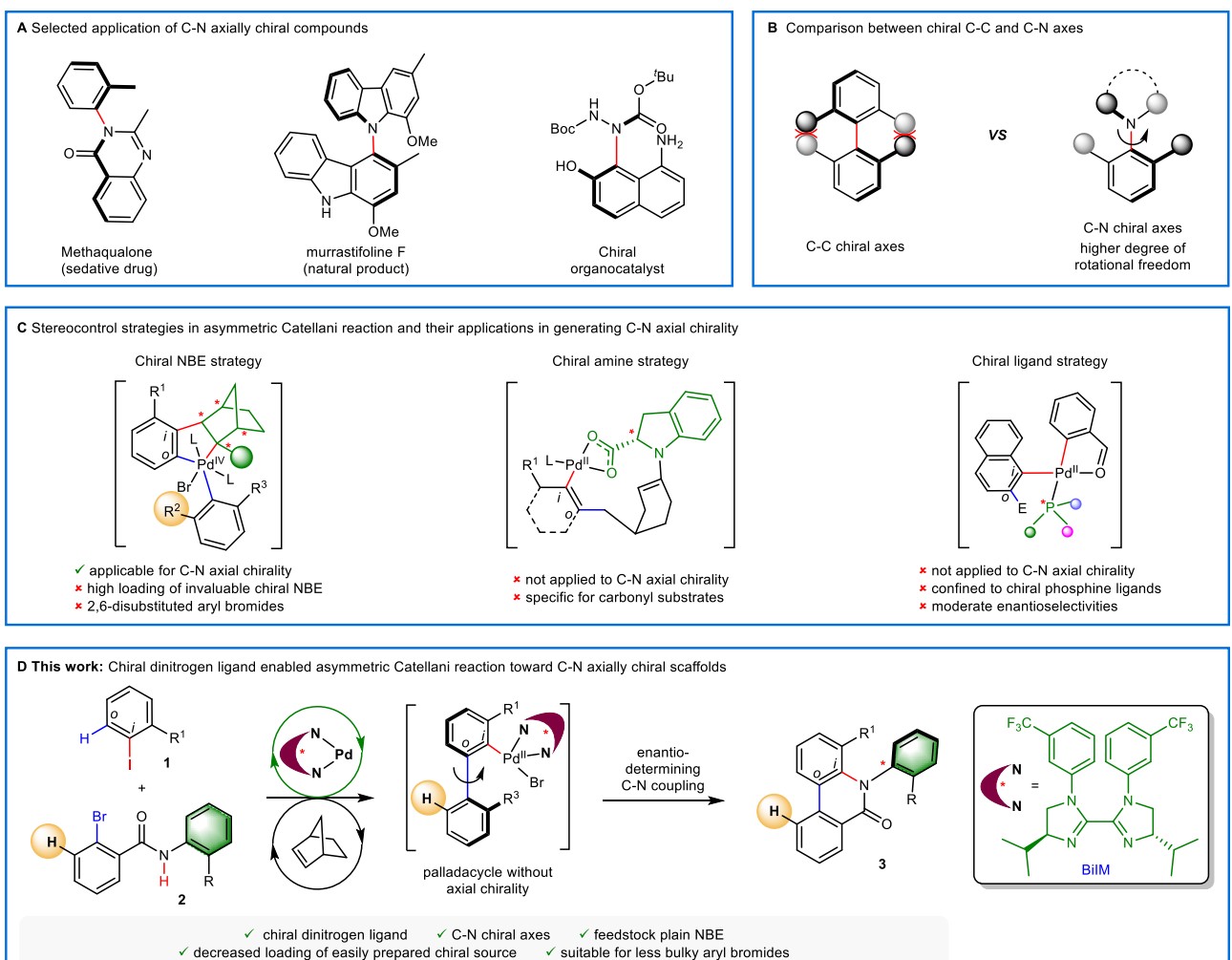

**Fig. 1 | Preparation of C–N axially chiral compounds via asymmetric Catellani approaches. A** Selected application of C–N axially chiral compounds. **B** Comparison between chiral C–C and C–N axes. **C** Stereocontrol strategies in asymmetric Catellani reaction and their applications in generating C–N axial chirality. **D** This work: chiral dinitrogen ligand enabled asymmetric Catellani reaction toward C–N axially chiral scaffolds. *i* = ipso, *o* = ortho.

amine[46,47] and chiral phosphine ligands[48,49] as the chiral source, respectively. However, the assembly of C–N axially chiral scaffolds through asymmetric Catellani approaches remains underdeveloped (Fig. 1C). In 2021, the Zhou group[50,51] established the generation of C–N axial chirality using chiral NBE strategy, a powerful asymmetric Catellani platform pioneered by Dong[38] and Zhou[39,41–44]. Despite the elegance of this method, the requirement of using high loading of less facilely accessible chiral NBE (typically 50 mol%) and bulky 2,6-disubstituted aryl bromides as coupling partners has significantly hampered the synthetic utility of this strategy. Considering the high loading of NBE in many of the Catellani reactions, sometimes even in super-stoichiometric amount[32–36,45], we postulated that the former could be addressed through the employment of an inexpensive NBE. However, the latter, originated from the indirect stereocontrol mode of chiral NBE strategy (generating a C–C chiral axis before the release of chiral NBE, followed by chirality transferring C–N formation)[50], could only be solved through tuning the chiral induction strategy. While chiral amine strategy developed by Zhou[46] and Gong[47] is specific for carbonyl coupling partners, the chiral ligand strategy pioneered by Gu[48,49] has only been applied to C–C formation involved asymmetric Catellani reactions in less satisfactory enantioselectivities[52]. In accordance with our continuous efforts on constructing C–N axially chiral scaffolds[12,26], we intended to explore a practical asymmetric Catellani route toward such compounds

complementary to chiral NBE strategy. Inspired by Gu's pioneering work[48,49] and the power of chiral ligands in related Pd-catalysis such as Pd(0) catalyzed asymmetric C–H activation[53–55], we turned our attention to the potentially general chiral ligand strategy. We envisioned that the reactivity toward C–N forming Catellani reaction, along with the enhancement of enantioselectivity could be achieved through judicious choice of chiral ligands. Importantly, the strategy holds great promise to reduce the loading of the chiral source when proper ligand sphere was identified.

Herein, we report a chiral dinitrogen ligand enabled asymmetric Catellani reaction, which allows the efficient and modular assembly of C–N axially chiral scaffolds from readily available starting materials (Fig. 1D). The chiral source (BiIM ligand), used in decreased loading when compared with chiral NBE strategy (20 mol% vs 50 mol %), could be readily prepared from cost-effective chiral amino alcohol, oxalate and aniline with a single-step column chromatography[56,57]. Less bulky mono-*ortho*-substituted bromoarenes, a broad range of coupling partners challenging to control the enantioselectivity with chiral NBE strategy[44], is compatible with our method. This is ascribed to the replacement of the indirect stereocontrol mode in chiral NBE strategy with a direct enantiodetermining C–N forming termination. To note, plain NBE, an inexpensive feedstock rarely been adopted in asymmetric Cetellani reactions, has been used as the mediator. The protocol is also featured with high

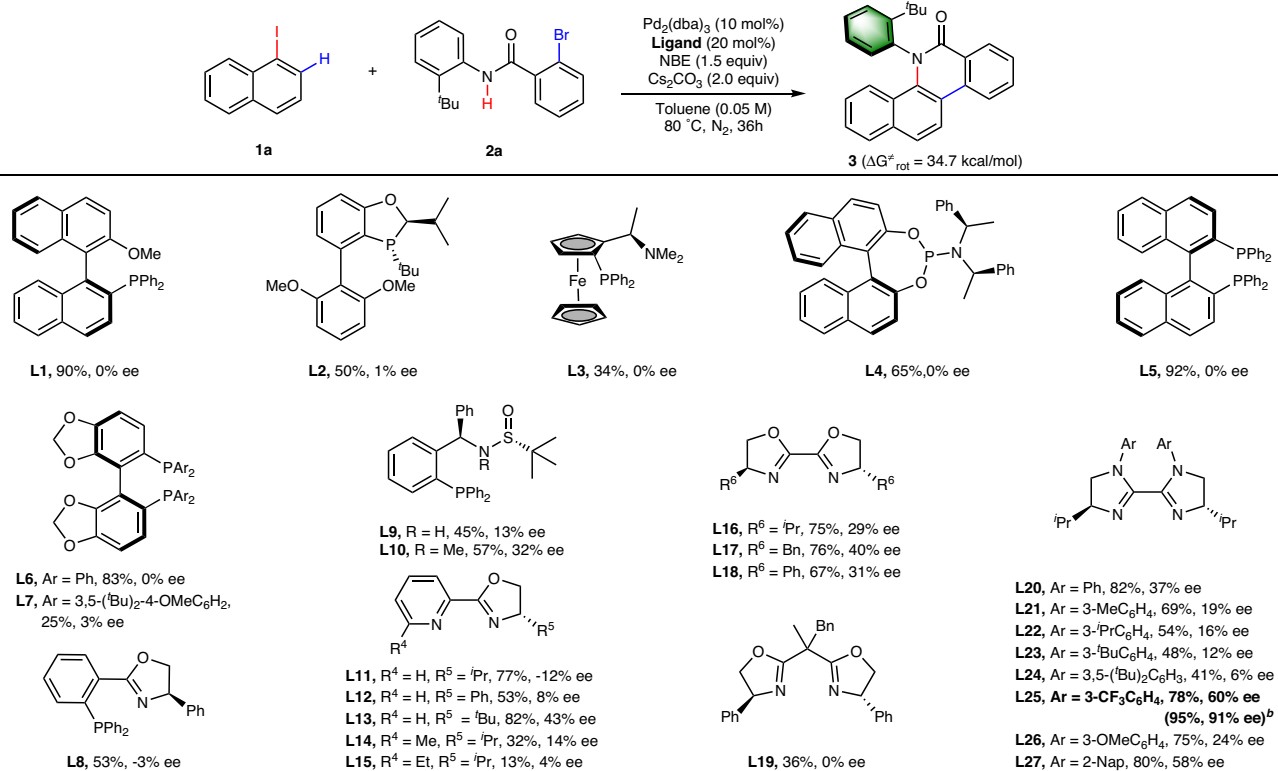

**Fig. 2 | Evaluation of chiral ligands.** [a]Reactions were carried out in 0.1 mmol scale with respect to amide **2**. Conditions: **1a** (1.5 equiv), **2a** (0.10 mmol), Pd₂(dba)₃ (5 mol%), Ligand (20 mol%), NBE (1.5 equiv), Cs₂CO₃ (2.0 equiv) in toluene (0.05 M) at 80 °C for 36 h under N₂. [b]Carried out under modified conditions: **1a** (1.5 equiv), **2a** (0.10 mmol), Pd₂(dba)₃ (5 mol%), **L25** (20 mol%), NBE (1.5 equiv), Ag₂SO₄ (2.0 equiv), H₂O (200 μL), 4 ÅMS (100 mg) in toluene (0.05 M) at 80 °C for 36 h under N₂. Yields were isolated yields and the ee's were determined by chiral HPLC analysis. Nap = naphthyl.

reactivity, good enantioselectivity, broad substrate scope, ease of operation and scale-up. The method has further enabled two-fold asymmetric Catellani reaction, affording chiral optoelectronic material candidates featuring two C–N chiral axes. Facile transformation of a product allows the stereospecific preparation of C–N axially chiral carboxylic acids (CCA), a potentially good chiral ligand for asymmetric C–H activation. This work significantly enriched the application of chiral ligand strategy, unveiling its potential of being a generally applicable asymmetric Catellani reaction platform complementary to chiral NBE strategy.

## Results

### Optimization of the reaction conditions

Initially, we selected 1-iodonaphthalene (**1a**) as the model substrate and 2-bromo-*N*-(2-(tert-butyl)phenyl)benzamide (**2a**) as the coupling partner. We commenced our investigation by screening various chiral ligands in the presence of Pd₂(dba)₃, plain NBE mediator, Cs₂CO₃ in toluene (Fig. 2). Although two chiral sulfonamide phosphine ligands (**L9**-**L10**) provided product **3** with noticeable enantioselectivity (13% and 32% ee), other monodentate or bidentate chiral phosphine ligands (**L1**-**L8**) only resulted in negligible enantioselectivity. In accordance with our hypothesis, these results highlight the challenge of using chiral phosphine ligands to construct of C–N chiral axes. We were pleased to see that chiral bidentate *N,N*-ligands forming 5-membered palladacycle upon chelating to a Pd-center, such as pyridinyl oxazoline ligands (**L11**-**L15**), BOX (**L16**-**L18**) and *N,N′*-aryl substituted biimidazoline (BiIM) ligands (**L20**-**L21**), afforded the desired product in promising enantioselectivity. Surprisingly, no enantioselectivity was observed with a BOX ligand that forms 6-membered palladacycle (**L19**). Upon further modulating the electronic properties and the bite angle of BiIM

ligands by varying the *N,N′*-substituents, we finally figured out **L25** bearing 3-trifluoromethylphenyl groups on both sp³ hybridized nitrogen atoms as the optimal ligand, affording the enantioenriched product in 78% yield and 60% ee. Through examining other reaction parameters (Supplementary Table 1 in the Supplementary Information), the optimal conditions were established when Cs₂CO₃ was replaced with Ag₂SO₄ and 200 μL of water and 100 mg of 4 Å molecular seivies were employed as additives, affording product **3** in 95% yield and 91% ee. The pivotal role of the catalyst, ligand and other reagents was confirmed by control experiments (Supplementary Table 3 in the Supplementary Information). We postulate that water improves the enantioselectivity by forming hydrogen bonding with the ligand and influencing the bite angle[58], whereas the silver salt serves as halide scavenger to improve the reactivity[59]. By heating product **3** in isopropanol at 150 °C and monitor the ee values, the rotational barrier was measured to be 34.7 kcal/mol, suggesting the remarkably high stability of this product even at 150 °C (t₁/₂(150 °C) = 8.48 hours, see Page 95–96 in the Supplementary Information for details).

### Substrate Scope

With the optimized condition in hand, we next investigated the scope of aryl iodides (Fig. 3). A series of *ortho*-substituted aryl iodides, such as 1-iodonaphthalene derivatives (**4**–**8**), 2-methyl iodobenzene (**9**–**10**), and 2-iodo-1,1′-biphenyl (**11**), are compatible with this method. Heteroarenes like dibenzo[b,d]furan (**12**) and the strongly coordinating quinoline moiety (**13**–**14**) are also viable substrates, affording the desired products in moderate to good yields and high enantioselectivities. The compatibility with diverse functional groups were tested on the 1-iodonaphthalene skeleton. In addition to the direct attachment of alkyl groups (**4**–**5**), ether (**6**) and

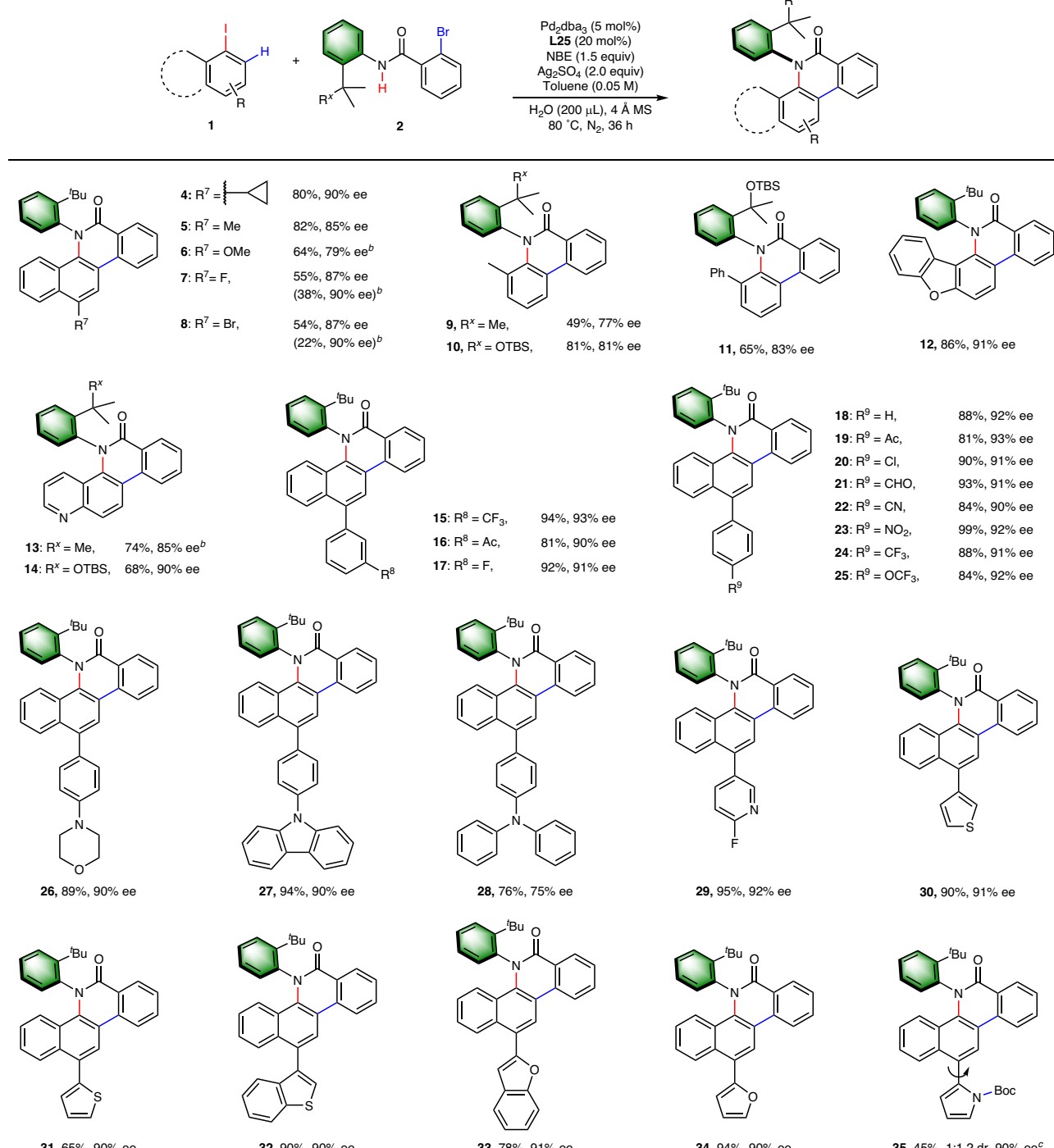

**Fig. 3 | Scope of Aryl Iodides.** [a]Reactions were carried out in 0.1 mmol scale with respect to amide **2**. Conditions: **1** (1.5 equiv), **2a** (0.10 mmol), Pd$_2$(dba)$_3$ (5 mol%), **L25** (20 mol%), NBE (1.5 equiv), Ag$_2$SO$_4$ (2.0 equiv), H$_2$O (200 μL), 4 ÅMS (100 mg) in toluene (0.05 M) at 80 °C for 36 h under N$_2$. [b]Dichloroethane instead of toluene was used as solvent. [c]The dr was observed through [1]H NMR, which disappeared after Boc-deprotection. The ee's were determined by chiral HPLC analysis. Ac = acetyl; TBS = *tert*-butyldimethylsilyl.

halides (**7–8**), both strongly electron-withdrawing and electron-donating groups are well tolerated when linked onto 1-iodonaphthalene through an aryl bridge (**15–28**). The direct attachment of heteroarenes, such as pyridine (**29**), thiophene (**30–32**), and furan (**33–34**), were also well tolerated. Enantioenriched product **35** (90% ee) was obtained as a mixture of rotamers in 1:1.2 ratio (detected by [1]H NMR), and the ratio disappeared after cleavage of Boc group. The yield of **35** was only moderate, perhaps due to the bulkiness of the protecting group.

Next, we further investigated the scope of the coupling partners (Fig. 4). *Ortho*-bromo benzamides bearing an electron-withdrawing and electron-donating groups on the benzoyl moiety are well-tolerated, affording the products in moderate to good yield and high enantioselectivity (**36–39**). Additionally, disubstituted bromo benzamides are also compatible (**41–42, 47–48**). Substitution on the aniline moiety was also allowed, with alkynyl (**44**), aryl (**46**) and alkyl (**45**) groups well tolerated. Notably, an additional bromine group *para* to the amine group (**43**) also afforded the target product in

**Fig. 4 | Scope of Amides.** [a]Reactions were carried out in 0.1 mmol scale with respect to amide **2**. Conditions: **1** (1.5 equiv), **2a** (0.10 mmol), Pd$_2$(dba)$_3$ (5 mol%), **L25** (20 mol%), NBE (1.5 equiv), Ag$_2$SO$_4$ (2.0 equiv), H$_2$O (200 μL), 4 ÅMS (100 mg) in toluene (0.05 M) at 80 °C for 36 h under N$_2$. [b]Dichloroethane instead of toluene was used as solvent. The ee's were determined by chiral HPLC analysis.

moderate enantioselectivity, regardless of its sterically more favourable nature over the bromine *ortho* to the benzoyl moiety. The *ortho tert*-butyl group at the aniline motif could be replaced by bulky alkoxyl groups (**49–50**), but the enantioselectivity is diminished when the coordinative hydroxyl group is not protected (**51**).

## Synthetic Application
Having investigated the scope of this protocol, a series of experiments were carried out to illustrate its synthetic utility (Fig. 5). Recently, the preparation of molecules possessing multiple chiral axes has attracted significant interest among organic chemistry community[60–64], for the unique role of these topologically complex scaffolds in material science and catalysis. In addition, the quick expansion of conjugate system during the reaction process further encouraged us to wonder whether products bearing more extended π-conjugate system were accessible. To our delight, the 1,5-diiodo-naphthalene substrate underwent two-fold asymmetric Catellani-type reaction smoothly and afforded products bearing two chiral C–N axes and a larger conjugate system (**52–54**). In accordance with the Horeau principle[65], ee of the chiral products amplified after two enantioselective operation (obtained in 94% to 99% ee). It appears that the ratio of meso-product raised as the size of bulky ortho group at the aniline motif increases (ranging from 10.8:1 to 1.4:1), likely due to the decreased repulsion within the meso-product, whose two bulky groups are located at different sides of the π-system. A gram-scale preparation of product **49** was carried out with equally high yield and ee (81% yield, 91% ee for gram-scale *vs* 83% yield and 92% ee for 0.1 mmol scale). Subsequent transformations,

such as cleavage of silyl group (**51**), protection of the free alcohol (**55**, CCDC 2249376), and oxidation of the alcohol to aldehyde (**56**), were all carried out with full retention of configuration. The enantioenriched C–N axially chiral aldehyde (**56**) could be further transformed into the corresponding oxime (**57**) and carboxylic acid (**58**) with high fidelity of enantiopurity. Considering the versatility of chiral carboxylic acid (CCA) ligands in asymmetric C–H activation, we subjected the obtained C–N axially chiral CCA (**58**) in a Ru(II)-catalysed enantioselective C–H annulation reaction[66,67]. To our delight, the reaction of sulfoximine **59** with sulfoxonium ylide **60** was carried out in high yield with moderate enantioselectivity under non-optimal conditions (96%, 55% ee). This application showcased the utility of our protocol in synthetic chemistry.

After obtaining an array of C–N axially chiral scaffolds bearing extended rigid-planar π-conjugate systems, we set out to examine the photophysical properties of selected products (Fig. 6). Photo luminescence quantum yields (PLQY) of these compounds are remarkably high, ranging from 40% to 97% (Fig. 6A). The fluorescence of these products ranges from bright blue to violet under UV light irradiation (Fig. 6B). Not surprisingly, the incorporation of an additional C–N stereogenic axis and more extended π-conjugate system has led to a red-shift in their absorption maxima (Fig. 6C, λ$_{abs}$ = 299 nm for **52**, **53** and **54** vs λ$_{abs}$ = 280 nm for **21** and **33**). The wavelength of emission maxima for **52**, **53** and **54** (λ$_{em}$ = 420 nm) is shorter than **21** (λ$_{em}$ = 445 nm) and **33** (λ$_{em}$ = 469 nm) (Fig. 6D). We further investigated the chiroptical properties of products **52**, **53**, **54** and their enantiomers by measuring their circular polarized luminescence (CPL) spectroscopies (Fig. 6E). Interestingly, all the selected three compounds are CPL-active, and each pair of

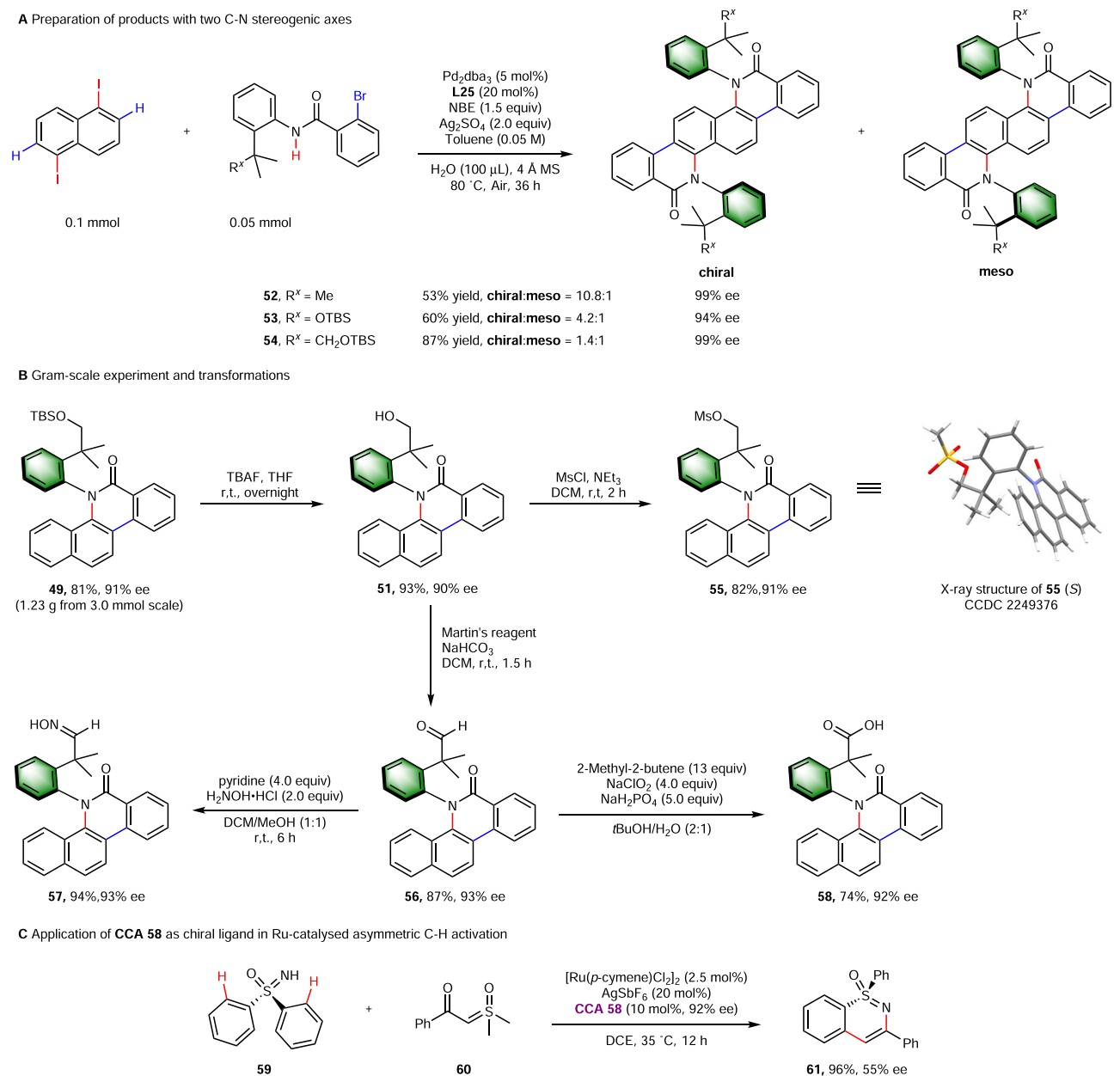

**Fig. 5 | Synthetic application. A** Preparation of products with two C–N stereogenic axes. **B** Gram-scale experiment and follow-up transformations. **C** Application of **CCA 58** as chiral ligand in Ru-catalysed asymmetric C–H activation. Ms = methanesulfonyl.

enantiomers displayed clear mirror images. The dissymmetry factors ($g_{lum}$) of both enantiomers of **52, 53, 54** around emission maxima (average value between 420–450 nm) range from $-9.8 \times 10^{-5}$ to $2.2 \times 10^{-4}$ (Fig. 6F), demonstrating the potential application of these easily accessible chiral compounds in developing interesting chiroptical devices[68].

## Discussion

We have developed a chiral dinitrogen ligand enabled asymmetric Catellani reaction that allows the efficient and modular assembly of C–N axially chiral scaffolds. Key to success of this methods is the utilization of an easily prepared and diversified chiral biimidazoline (BiIM) ligand that possesses a controllable bite angle as well as variable electronic and steric properties. The method features broad substrate scope, good reactivity, high enantioselectivity and ease of scale-up. A series of transformations around one of the products (**49**) were all carried out with full retention of configuration. A C–N

axially chiral carboxylic acid (**CCA 58**) derived from **49** has showcased its potential as the chiral ligand for Ru-catalysed asymmetric C–H activation. Two-fold Catellani reaction with a diiodoarene afforded several CPL-active compounds bearing two C–N chiral axes in high enantioselectivity, which are potential candidates for chiroptical materials. The successful enantiocontrol in the construction of C–N axial chirality with chiral dinitrogen ligand strongly suggest that chiral ligand strategy has the potential to become a generally applicable asymmetric Catellani platform complementary to chiral NBE strategy. Further application of this system in other Catellani-type reactions is ongoing in our lab.

## Methods

### General procedure for BiIM ligand enabled asymmetric Catellani reaction

To an oven-dried 10 mL Schlenk tube were added substrate **1** (0.15 mmol), amide **2** (0.1 mmol), Pd$_2$(dba)$_3$ (4.9 mg, 0.005 mmol), **L25**

**A** Structure and PLQY information of selected compounds for photophysical property characterization

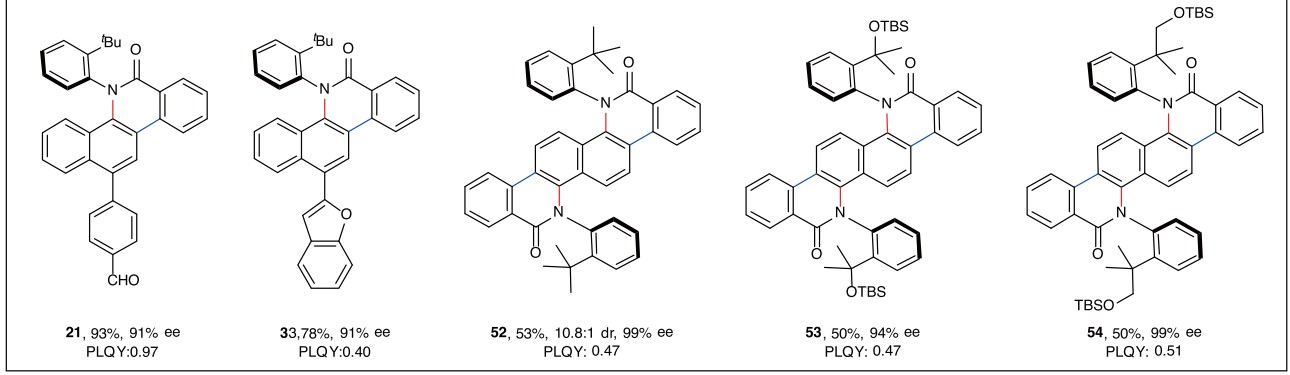

**21**, 93%, 91% ee
PLQY:0.97

**33**,78%, 91% ee
PLQY:0.40

**52**, 53%, 10.8:1 dr, 99% ee
PLQY: 0.47

**53**, 50%, 94% ee
PLQY: 0.47

**54**, 50%, 99% ee
PLQY: 0.51

**B** Fluorescence images of selected compounds ( $\lambda_{ex}$= 365 nm)

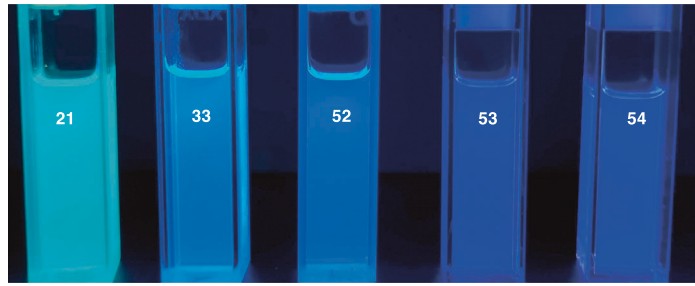

**C** Absorption spectra in DCM (20 μM)

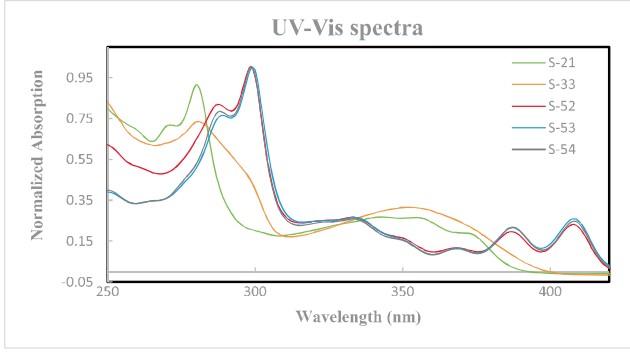

**D** Emission spectra in DCM (20 μM)

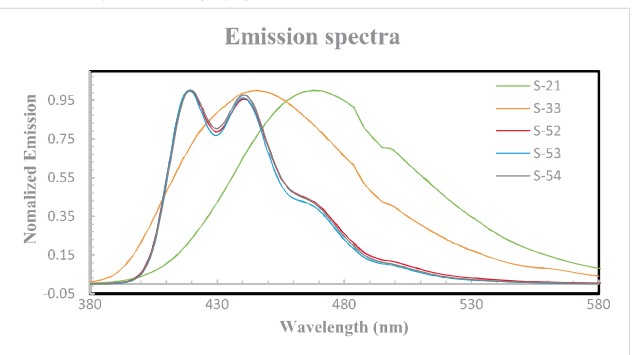

**E** CPL spectra in DCM (1.0 mM, $\lambda_{ex}$= 365 nm)

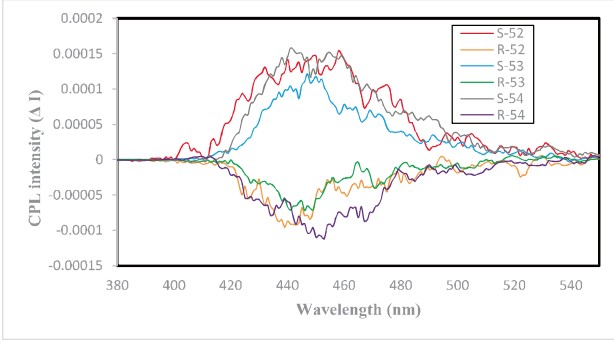

**F** $g_{lum}$ values-wavelength curves

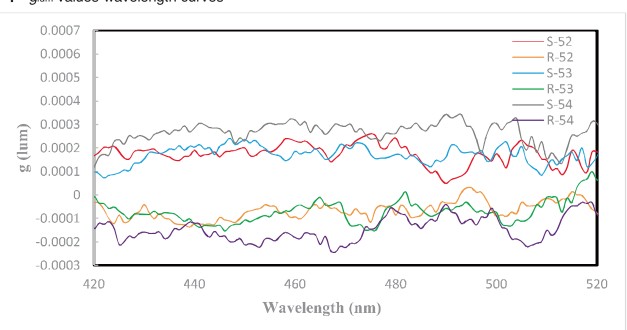

**Fig. 6 | Photophysical Property Investigations. A** Structure and PLQY information of selected compounds for photophysical property characterization. **B** Fluorescence images of selected silicon-bridged heterocycles ($\lambda_{ex}$ = 365 nm). **C** Absorption spectra of selected compounds in DCM (20 μM). **D** Emission spectra of selected compounds in DCM (20 μM). **E** CPL spectra of compounds in DCM (1.0 mM) at room temperature, excited at 300 nm. **F** $g_{lum}$ values-wavelength curves.

(10.2 mg, 0.020 mmol), Ag$_2$SO$_4$ (62.3 mg, 0.2 mmol), Toluene or 1,2-dichloroethane (2.0 mL), H$_2$O (0.2 mL), 4ÅMS (100 mg), NBE (15 mg, 0.15 mmol). The mixture was stirred for 36 h at 80 °C. The resulting mixture was quenched by filtered through a celite pad and concentrated in *vacuo*. The residue was purified by preparative TLC to afford the product.

## Data availability
Experimental procedures and characterization data are available within this article and its Supplementary Information. All other data are available from the corresponding author upon request. The X-ray crystallographic coordinates for the structure of compound **55** reported in this study have been deposited at the Cambridge

Crystallographic Data Centre (CCDC), under deposition number 2249376. These data can be obtained free of charge from The Cambridge Crystallographic Data Centre via www.ccdc.cam.ac.uk/data_request/cif.

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

## Acknowledgements

Financial support from National Natural Science Foundation of China (21925109, U22A20388, 92256302, for B.-F.S.; 22201249 for Q.Z.), National Key R&D Program of China (2022YFA1504302, 2021YFF0701603 for B.-F.S.), Fundamental Research Funds for the Central Universities (226-2023-00115, 226-2022-00224), Zhejiang Provincial NSFC (LD22B030003 for B.-F.S.), the China Postdoctoral Science Foundation (2023M733044 for L.J.), and ZJU-Hangzhou Global Scientific and Technological Innovation Center are gratefully acknowledged.

## Author contributions

L.J., Y.L., and Y.M. contributed equally. L.J., Y.L. Q.Z. and B.-F.S. conceived the project. L.J., and Y.L. performed the majority of experiments and analysed the data. Y.M. and Z.L. provided most BiIM ligands and X.-B.H. participated in substrate preparation. Q.Z. and B.-F.S. directed the project and wrote the manuscript.

## Competing interests

L.J., Q.Z., and B.F.S. have filed a provisional patent application (202310716817.8). All other authors declare no competing interests.
