## [Peer Review File · Nature Communications]

Chiral dinitrogen ligand enabled asymmetric Pd/norbornene cooperative catalysis toward the assembly of C–N axially chiral scaffoldsREVIEWER COMMENTS

Reviewer #1 (Remarks to the Author):

Shi, Zhang and coworkers describe a catalytic access to phenanthridinones displaying a C-N chirality by means of an asymmetric version of the Catellani reactions that features the use of chiral dinitrogen ligands. In general, the preparation of compounds with C-N axial chirality is highly interesting among the organic chemistry community. The authors demonstrate the generality of the reaction and a satisfactory substrate scope is reported. However, I believe that, although the study shows the scientific soundness necessary for the journal, the generality of the reported transformations is limited to a specific scaffold (phenanthridinones), that was already synthesized under Catellani non-asymmetric conditions (Org. Lett. 2004, 6, 4759-4762). Consequently, the interest might be restricted to a narrow audience and for these reasons I suggest submission to a more specialized journal.

Reviewer #2 (Remarks to the Author):

[See attachments]

Reviewer #3 (Remarks to the Author):

In the presented manuscript, Zhang, Shi and colleagues established an interesting chiral BiIM ligands promoted asymmetric Catellani approach for the challenging preparation of invaluable C–N axially chiral compounds. The authors turned to chiral ligand strategy because in the construction of C–N chiral axes via Catellani route, only chiral NBE strategy is operative to date, but the existence of several inherent limitations originated from the chiral induction mode has significantly hampered its broad use. The limitations include the requirement of using invaluable chiral NBE in half stoichiometric amount, and the limitation of using bulky 2,6-disubstituted bromoarenes as the coupling partners. Indeed, through identification of a chiral BiIM ligands, these formidable challenges are successfully addressed. To the best of my knowledge, this is the first time that a chiral dinitrogen ligand has been used as the chiral source of an asymmetric Catellani reaction. Contrary to the less facile preparation of chiral NBE, BiIM ligands involved in this work are easily prepared and diversified, and the employed NBE is an expensive feedstock. The protocol also features broad substrate scope, high reactivity, good enantioselectivity, ease of operation and scale-up. Application of this method in the preparation of chiral optoelectronic material candidates featuring two C–N chiral axes and a chiral ligand for asymmetric C–H activation has also been demonstrated, highlighting the synthetic utility of this method. Further, in the previously reported asymmetric Catellani reactions based on chiral ligand strategy, only chiral phosphine ligands are adopted, forging C–C bonds in moderate enantioselectivity. The unprecedented employment of chiral dinitrogen ligands for the construction of C–N chiral axes in this work would significantly enriched the application of chiral ligand strategy, unveiling its potential of being a second generation of generally applicable asymmetric Catellani reaction platform.

Taken together, I strongly support the acceptance of this manuscript in Nature Communications after answering the questions/revisions listed below.

1. It is interesting that the addition of water could significantly improve the enantioselectivity. Please provide a plausible comment on the effect of these additives.
2. According to previous reports, it is rare to use silver salts rather than base in an asymmetric Catellani reaction. Explain why is the reactivity significantly enhanced with silver salts.
3. For product 3 bearing an ortho-tert-butyl substituent on the anilide motif, the rotational barrier was measured to be 34.7 kcal/mol, which is high since the reaction temperature is only 80 °C. Is a smaller substituent bulky enough to preserve the C-N axial chirality?

This manuscript by Shi and co-workers describes a convergent approach for the construction of C–N axial chirality based on a novel asymmetric Catellani-type reaction using a chiral biimidazole ligand. Although the Catellani reaction has advanced considerably since its discovery in the 1990s, exploitation of this asymmetric tactic has rarely been initiated for a long time. It is only until recently that remarkable breakthroughs have been achieved using chiral phosphine ligand, chiral amine or chiral norbornene co-catalyst as the stereoinduction element. As a typical example, the Zhou group recently developed an elegant strategy for the construction of C–N axial chirality via Pd/chiral norbornene cooperative catalysis and axial-to-axial chirality transfer (Refs. 50-51). In Zhou's work, chiral norbornene co-catalyst was used as the chiral source, and the obtained C–N axial chirality originates from the preformed transient C–C axial chirality. In stark contrast, the strategy utilized in this work for the construction of C–N axial chirality is significantly different with respect to both the chiral source and the stereoinduction model. To be specific, the chiral biimidazole ligand is used as the chiral source, and the C–N axial chirality is directly determined during the final C–N bond formation. Key features of this method include broad substrate scope, good enantioselectivity, scalability and ease of operation. The synthetic application and utility of the obtained C–N axially chiral phenanthridinones were demonstrated by the preparation of chiral optoelectronic materials and chiral ligands. In addition, this paper is well written and the authors complete a reasonable analysis of their observations. The supporting information is of good quality for the characterization of new compounds.

Overall, this study represents a nice advance in the emerging research area of asymmetric Catellani reaction and might bring great opportunity to make the use of chiral dinitrogen ligand become a generally applicable strategy in Pd/norbornene cooperative catalysis. Therefore, I support the publication of this manuscript in the esteemed *Nature Communications*, after the following minor issues being adequately addressed.

1. In the introduction part, "...hasn't met any success until very recently, by using chiral NBE³⁷⁻⁴⁵, chiral amine^{46,47} ..." Even though ref. 37 is one of the earliest reports that uses chiral NBE, it was used to promote functional group directed asymmetric meta-C-H activation, not the herein described asymmetric Catellani reaction initiated from aryl iodides. A note is needed to distinguish it from ref. 38-45.
2. A very recent work on the efforts to develop an asymmetric Catellani reaction with chiral ligands, although with low yield and enantioselectivity, should be cited (Chem. Sci., 2024, 15, 1318)
3. Figure 2, please unify the style of "t" and "i" in tBu and iPr groups, either superscript or italic.
4. All the compound numbers should be in bold, please check the main text and figure notes.
5. In previous work via chiral NBE strategy (ref. 50), a meta-substitution of the aryl iodide significantly increased the rotation barrier as compared to the same substituent at the ortho-position. I wonder if meta-substituted aryl iodides are viable substrates in this system?
6. Only one decimal is required for the characterization of ¹³C-NMR.

Responds to the reviewers' comments

Reviewer #1 (Remarks to the Author):

Comments: Shi, Zhang and coworkers describe a catalytic access to phenanthridinones displaying a C-N chirality by means of an asymmetric version of the Catellani reactions that features the use of chiral dinitrogen ligands. In general, the preparation of compounds with C-N axial chirality is highly interesting among the organic chemistry community. The authors demonstrate the generality of the reaction and a satisfactory substrate scope is reported. However, I believe that, although the study shows the scientific soundness necessary for the journal, the generality of the reported transformations is limited to a specific scaffold (phenanthridinones), that was already synthesized under Catellani non-asymmetric conditions (Org. Lett. 2004, 6, 4759-4762). Consequently, the interest might be restricted to a narrow audience and for these reasons I suggest submission to a more specialized journal.

Response: *We appreciate the reviewer for the positive comments on the synthetic goal of our work, "compounds with C-N axial chirality", "is highly interesting among the organic chemistry community". The reviewer also said our method has "a satisfactory substrate scope" and "the study shows the scientific soundness necessary for the journal". However, we are surprised at the final suggestion, namely to submit elsewhere, due to the limitation of scaffold type of the products, and the fact that the racemic edition has previously been reported.*

In my opinion, the racemic reports could not influence the novelty of our work because the discovery of novel scaffolds and the realization of chirality control belongs to different dimensions of novelty. The precise stereocontrol could not be achieved without careful design on the stereocontrol model and the type of chiral source, together with extensive screening on the precise structure of the chiral source and other reaction parameters. These challenges do not exist in racemic reactions that focus on discovering novel scaffolds or methods.

As for the limitation in product types, there are two possible reasons. First, to retain the chirality of the products conformationally stable both at the reaction temperature and ambient temperature, especially for atropochiralities such as C-N axial chirality in this work, the product type might be limited. Second, the structure of the chiral source, such as the chiral ligand used in this work, could significantly influence the electronic and steric properties of the catalyst, resulting in limitations in substrate scope. Such limitation in product types is common in asymmetric catalysis, as exemplified by the recent advancements on asymmetric Catellani reactions (ref. 40-51).

We believe that the novel aspects of this work, such as the unprecedented utilization of chiral dinitrogen ligands in asymmetric Catellani reactions, the enantiocontrol model distinct from previously established chiral NBE strategy, and the application in preparation of chiroptical material candidates would be interesting for broad readership among directions such as asymmetric synthesis, organometallic chemistry and material sciences. Our work might also encourage future endeavors of using chiral dinitrogen ligand strategy to achieve asymmetric Pd/norbornene cooperative catalysis, leading to a generally applicable asymmetric platform with less limitations in the product type. We truly hope the reviewer to rethink about the novelty of this

manuscript.

Reviewer #2 (Remarks to the Author):

Comment 1: In the introduction part, "...hasn't met any success until very recently, by using chiral NBE³⁷⁻⁴⁵, chiral amine^{46,47} ..." Even though ref. 37 is one of the earliest reports that uses chiral NBE, it was used to promote functional group directed asymmetric meta-C-H activation, not the herein described asymmetric Catellani reaction initiated from aryl iodides. A note is needed to distinguish it from ref. 38-45.

Response: *We thank the reviewer for the nice suggestion. In our revised manuscript, we added a note that read "by using chiral NBE (previously designed for functional group directed remote meta-C-H activation³⁷)³⁸⁻⁴⁵," so that ref. 37 is distinguished from ref. 38-45.*

Comment 2: A very recent work on the efforts to develop an asymmetric Catellani reaction with chiral ligands, although with low yield and enantioselectivity, should be cited (Chem. Sci., 2024, 15, 1318).

Response: *We truly appreciate this professional suggestion. The suggested reference is inserted in our revised manuscript as ref. 52, and reference numbers after it have been changed accordingly.*

Comment 3: Figure 2, please unify the style of "t" and "i" in tBu and iPr groups, either superscript or italic.

Response: *We thank the reviewer for this kind suggestion. According to other figures, we chose to express such "t" and "i" in both superscript and italic. Corresponding corrections are made in Figure 2 and the supplementary information in our revision.*

Comment 4: All the compound numbers should be in bold, please check the main text and figure notes.

Response: *We thank the reviewer for this kind suggestion. We checked throughout the manuscript and revised accordingly at the notes for Fig. 3, Fig. 4 and page 9 in our revised manuscript.*

Comment 5: In previous work via chiral NBE strategy (ref. 50), a *meta*-substitution of the aryl iodide significantly increased the rotation barrier as compared to the same substituent at the ortho position. I wonder if *meta*-substituted aryl iodides are viable substrates in this system?

Response: *This is indeed a good suggestion. Based on the scope of meta-substituted aryl iodides in ref. 50, we tested three aryl iodides with chloro, trifluoromethyl or ester group at the meta-position under our standard conditions. However, no target product was isolated. Instead, the major product resulted from the coupling between an aryl iodide, a norbornene and two aryl bromides has been isolated (see the figure below). The reason why we cannot obtain the target products with meta-substituted aryl iodides is probably **due to the use of plain NBE mediator without bulky***

substituents, which hampered the release of NBE via β -C elimination. This will be discussed in details in the proposed mechanism below.

The structure is deduced based on our $^1\text{H-NMR}$, $^{13}\text{C-NMR}$ and HRMS characterization, the reaction mechanism and comparison with related references. The first strong evidence for such a structure is the significant upfield shift for one of the tert-butyl groups in $^1\text{H-NMR}$ (one is around 1.35 ppm, and the other is around 0.78 ppm), which suggest that the tert-butyl group locates at the shielding area of aryl-norbornane scaffolds, most likely via C–N coupling between the anilide motif and unreleased NBE (for analogues with such significant shifts, see product **5i** in ACS Catal. 2021, 11, 14, 8585; J. Org. Chem. **2018**, 83, 13930). Our HRMS results further suggest the dehydrobromination of an aryl bromide, likely via C–H activation/annulation.

A plausible mechanism for the generation of such products was also proposed (see the following figure). First, similar to a normal Catellani reaction, a norbornene mediated C–H arylation occurred at the less hindered ortho-position of the aryl iodide in the presence of Pd(0) catalyst. But the resulted intermediate **I** could not proceed β -C elimination to release the NBE. This is because intermediate **I** generated from meta-aryl iodide and plain NBE **lacks ortho-disubstitution on the aryl ring** (Angew. Chem. Int. Ed. Engl. **1995**, 33, 2421) **and bulky group on the NBE skeleton** (Nat. Chem. **2018**, 10, 866) that's essential for β -C elimination. More specifically, in the Nat. Chem. paper, different NBEs were evaluated to test whether mono-ortho substitution is enough for β -C elimination of NBE in the case of meta-substituted aryl iodides, resulting in mono-ortho amination product (supplementary information, page 5, Supplementary Table 2). While the methyl ester analogue (**N9**) of the chiral NBE used in ref. 50 (in ethyl ester form) could afford mono-ortho amination product in over 20% yield, plain NBE (**N1**) used in our work only afforded trace amount of such product. This explains why meta-aryl iodides are viable substrates in ref. 50, but not in our system. Due to the large size of our aryl bromide, a second arylation at the more hindered ortho-position is not likely to proceed. Additionally, without the release of NBE, intramolecular C–N coupling from intermediate **I** is also not favorable due to the increased ring size. Instead, ligand exchange with another molecule of **2a** is more plausible, followed by intramolecular oxidative addition to generate Pd(IV) species **III**. Subsequently, a C–N forming reductive elimination afforded Pd(II) species **IV**, which proceed C–H activation to generate a seven-membered palladacycle. Finally, a reductive elimination affords a new C–C bond and regenerates the Pd(0) catalyst.

The tested meta-substituted aryl iodides are summarized in a Figure S7 as “unsuccessful substrates” in our revised supplementary information. For the characterizations we did for these products, please refer to our attached data at the end of this file for details.

Comment 6: Only one decimal is required for the characterization of ¹³C-NMR.

Response: We thank the reviewer for this kind suggestion. The ¹³C-NMR data has been corrected to one decimal places in our revised supplementary information.

Reviewer #3 (Remarks to the Author):

Comment 1: It is interesting that the addition of water could significantly improve the enantioselectivity. Please provide a plausible comment on the effect of these additives.

Response: We thank the reviewer for this insightful suggestion. Very recently, a review that summarize the role of water in asymmetric catalysis has been reported (*Org. Biomol. Chem.* **2024**, *22*, 2510), which suggests that water can influence the reaction outcome (including enantioselectivity) via effects such as hydrophobic, hydrogen bonding, protonation. Considering the results in our optimization table of the reaction conditions (page 19 of the supplementary information, Table S1), we hypothesis that water might have two roles in our system. First, when an inorganic base is used, the addition of a small amount of water is essential for achieving enantioselectivity (entries 1-5). We think that **water might form hydrogen bond with the BiIM**

ligand, so that the bite angle of ligand is tuned to facilitate a more precise stereocontrol. Second, when a silver salt instead of an inorganic base is used, the amount of water should be significantly increased (entries 6-9). According to silver-catalyzed asymmetric catalysis (i.g. *J. Am. Chem. Soc.* **2007**, 129, 750), it is possible that a small amount of silver salt could compete with palladium catalyst to coordinate with the chiral ligand. It is also possible that Ag^+ might influence the bite angle of ligand originally optimized by water, leading to decreased enantioselectivity. We postulated that **more water is required to solvate the silver salt, so that the competing with Pd-catalyst is reduced, and the bite angle of ligand is preserved.** These proposed effects are depicted in the following figure.

We also added a simple explanation in our revised manuscript, which read “We postulate that water improves the enantioselectivity by forming hydrogen bonding with the ligand and influencing the bite angle⁵⁸”. Ref. 58 refers to the review noted above (*Org. Biomol. Chem.* **2024**, 22, 2510).

Proposed roles of water and silver salts in enantiocontrol

Comment 2: According to previous reports, it is rare to use silver salts rather than base in an asymmetric Catellani reaction. Explain why is the reactivity significantly enhanced with silver salts.

Response: This is a good point. Actually, the use of silver salts instead of inorganic base is rare in both asymmetric and racemic Catellani reactions. However, the Dong group recently reported the racemic Catellani-type reaction of electronrich heteroarenes, in which silver salts are essential for the high reactivity (*J. Am. Chem. Soc.* **2019**, 141, 18958; *Angew. Chem. Int. Ed.* **2021**, 60, 26184). Similar to their hypothesis, we think that **the silver salts might act as halide scavenger in our system, so that the reactivity is significantly enhanced.** The use of silver salts is more common in Pd-catalyzed C–H activation, a process also involved in the ortho-functionalization stage of Catellani reactions. In addition to acting as halide scavenger, silver salts might also play other roles such as oxidants or generating important Pd-Ag bimetallic complex that lowers the energy barriers (*J. Organomet. Chem.* **2018**, 864, 19-25) in these reactions. We cannot rule out these complicated roles at this stage.

We also added a simple explanation in our revised manuscript, which read “whereas the silver salt serves as halide scavenger to improve the reactivity⁵⁹”. Ref. 59 refers to a paper noted above (*Angew. Chem. Int. Ed.* **2021**, 60, 26184), since it clearly claimed the possible role of silver salt.

Comment 3: For product 3 bearing an *ortho-tert*-butyl substituent on the anilide motif, the rotational barrier was measured to be 34.7 kcal/mol, which is high since the reaction temperature is only 80 °C. Is a smaller substituent bulky enough to preserve the C-N axial chirality?

Thank you for the suggestion. We tried to replace the *tert*-butyl group on the anilide motif with smaller *iso*-propyl and benzo-fused ring, only to afford racemic results (*iso*-propyl: 43% yield, 0% ee; benzo-fused ring: 54% yield, 0% ee). This is in accordance with ref. 50, where anilides with smaller substituents only afforded negligible results even in the presence of an additional methyl group (Figure 4 in ref. 50, compounds **3N**, **3O** and **3P**). In their further investigation, enantioenriched **3N** (70% ee) generated from deprotection of **3I** (92% ee) is totally racemized upon heating at 70 °C for 3 hours, due to the decreased rotational barrier of C–N chiral axis. Considering the higher reaction temperature and longer reaction time in our standard conditions (80 °C, 36 h), *it is most likely that we got racemic results because the rotational barrier of C–N chiral axis is significantly decreased in the case of smaller substituents.*

Data for the aryl bromides (numbered as **2p** and **2q**) and product (numbered as **64** and **65**) was added where appropriate in the revised supplementary information.

Supplementary data for comment 5 from reviewer 2

2'-(3-(4-(*tert*-butyl)-6-oxophenanthridin-5(6H)-yl)bicyclo[2.2.1]heptan-2-yl)-N-(2-(*tert*-butyl)phenyl)-4'-chloro-[1,1'-biphenyl]-2-carboxamide

The compound was prepared according to **General Procedure C** (for details of this procedure, please refer to our supplementary information) and purified by preparative TLC in hexane/EtOAc = 6/1 as the eluent to afford a colorless oil (10 mg, 28% yield).

¹H NMR (600 MHz, Chloroform-*d*) δ 8.24 (d, $J = 7.8$ Hz, 1H), 8.11 (s, 1H), 8.04 (d, $J = 8.2$ Hz, 1H), 7.99 (d, $J = 2.4$ Hz, 1H), 7.89 – 7.83 (m, 1H), 7.70 – 7.64 (m, 1H), 7.59 – 7.53 (m, 1H), 7.50 – 7.44 (m, 2H), 7.44 – 7.39 (m, 2H), 7.39 – 7.31 (m, 3H), 7.16 (q, $J = 7.1$ Hz, 2H), 6.90 – 6.80 (m, 2H), 6.52 (d, $J = 7.7$ Hz, 1H), 3.52 (d, $J = 9.6$ Hz, 1H), 2.59 (d, $J = 9.7$ Hz, 1H), 2.29 (dd, $J = 25.8, 4.1$ Hz, 2H), 2.13 (d, $J = 10.6$ Hz, 1H), 1.56 – 1.46 (m, 1H), 1.37 (m, 11H), 1.24 – 1.18 (m, 1H), 0.78 (s, 9H), 0.64 – 0.50 (m, 1H).

¹³C NMR (151 MHz, Chloroform-*d*) δ 167.8, 163.2, 146.1, 141.4, 141.4, 138.8, 138.2, 137.2, 135.7, 134.9, 133.1, 132.8, 130.6, 130.0, 129.2, 128.8, 128.8, 128.6, 128.4, 128.0, 127.1, 126.8, 126.6, 126.1, 126.0, 125.9, 125.8, 125.7, 122.8, 121.9, 120.4, 52.7, 49.2, 45.4, 41.5, 37.5, 35.6, 34.6, 31.3, 31.0, 30.5, 30.3.

HRMS (ESI-TOF) calcd for C₄₇H₄₈ClN₂O₂⁺ ([M+H]⁺): 707.3399, found: 707.3397.

¹H NMR (600 MHz, Chloroform-*d*)

¹³C NMR (151 MHz, Chloroform-*d*)

HRMS (ESI-TOF)

Sample Name
User Name
Sample Type
ACQ Method

Sample43
Sample
MSMS240326.m

Position P1-D10
Inj Vol 0.2
IRM Calibration Status Some Ions Missed
Comment

Instrument Name Instrument 1
InjPosition
Data Filename JL-C48-2-1-MSMS.d
Acquired Time 3/26/2024 4:00:10 PM (UTC+08:00)

2'-(3-(4-(tert-butyl)-6-oxophenanthridin-5(6H)-yl)bicyclo[2.2.1]heptan-2-yl)-N-(2-(tert-butyl)phenyl)-4'-(trifluoromethyl)-[1,1'-biphenyl]-2-carboxamide

The compound was prepared according to **General Procedure C** (for details of this procedure, please refer to our supplementary information) and purified by preparative TLC in hexane/EtOAc = 6/1 as the eluent to afford a colorless oil (16 mg, 43% yield).

¹H NMR (600 MHz, Chloroform-*d*) δ 8.28 (d, $J = 7.9$ Hz, 1H), 8.21 (s, 1H), 8.11 (d, $J = 8.2$ Hz, 1H), 8.06 (s, 1H), 7.98 – 7.89 (m, 1H), 7.70 (ddd, $J = 8.3, 7.1, 1.4$ Hz, 1H), 7.62 – 7.57 (m, 1H), 7.53 (s, 1H), 7.50 (t, $J = 7.5$ Hz, 1H), 7.43 (m, 2H), 7.35 – 7.29 (m, 3H), 7.13 (ddd, $J = 11.2, 6.4, 2.4$ Hz, 2H), 6.91 (t, $J = 7.6$ Hz, 1H), 6.89 – 6.84 (m, 1H), 6.66 (d, $J = 7.8$ Hz, 1H), 3.55 (d, $J = 9.6$ Hz, 1H), 2.90 (d, $J = 9.6$ Hz, 1H), 2.31 (dd, $J = 20.3, 3.6$ Hz, 2H), 2.19 (d, $J = 10.8$ Hz, 1H), 1.53 (tt, $J = 12.3, 4.4$ Hz, 1H), 1.44 – 1.37 (m, 1H), 1.33 (m, 10H), 1.21 – 1.12 (m, 1H), 0.78 (s, 9H), 0.52 (td, $J = 10.4, 9.6, 3.0$ Hz, 1H).

¹³C NMR (151 MHz, Chloroform-*d*) δ 167.5, 163.5, 146.1, 142.3, 141.4, 141.0, 138.3, 136.9, 135.7, 135.1, 133.6, 133.3, 132.9, 130.2, 129.3, 129.1, 128.9, 128.6, 128.5, 127.4 (q, $J_{CF} = 3.0$ Hz), 127.1, 126.7, 126.6, 126.0, 125.9, 125.8, 125.7, 125.6, 124.4 (q, $J_{CF} = 271.8$ Hz), 123.7 (q, $J_{CF} = 33.2$ Hz), 122.0, 121.2, 117.8 (q, $J_{CF} = 4.5$ Hz), 52.7, 49.2, 44.6, 42.0, 37.9, 35.7, 34.5, 31.4, 30.9, 30.6, 30.2.

¹⁹F NMR (565 MHz, Chloroform-*d*) δ -61.9.

HRMS (ESI-TOF) calcd for $C_{48}H_{48}F_3N_2O_2^+$ ($[M+H]^+$): 741.3662, found: 741.3664.

¹H NMR (600 MHz, Chloroform-d)

¹³C NMR (151 MHz, Chloroform-d)

¹⁹F NMR (565 MHz, Chloroform-d)

pdata/1

HRMS (ESI-TOF)

Sample Name	Sample44	Position	P1-D11	Instrument Name	Instrument 1
User Name		Inj Vol	0.2	InjPosition	
Sample Type	Sample	IRM Calibration Status	All Ions Missed	Data Filename	JL-C48-3-1-MSMS.d
ACQ Method	MSMS240326.m	Comment		Acquired Time	3/26/2024 4:07:59 PM (UTC+08:00)

methyl 2-(3-(4-(tert-butyl)-6-oxophenanthridin-5(6H)-yl)bicyclo[2.2.1]heptan-2-yl)-2'-((2-(tert-butyl)phenyl)carbamoyl)-[1,1'-biphenyl]-4-carboxylate

The compound was prepared according to **General Procedure C** (for details of this procedure, please refer to our supplementary information) and purified by preparative TLC in hexane/EtOAc = 6/1 as the eluent to afford a colorless oil (17 mg, 47% yield).

¹H NMR (600 MHz, Chloroform-*d*) δ 8.71 (s, 1H), 8.25 (d, J = 8.0 Hz, 1H), 8.21 (d, J = 8.2 Hz, 1H), 8.07 (s, 2H), 7.92 – 7.82 (m, 1H), 7.70 (ddd, J = 8.3, 7.1, 1.4 Hz, 1H), 7.60 – 7.54 (m, 1H), 7.48 (t, J = 7.5 Hz, 1H), 7.46 – 7.40 (m, 2H), 7.39 (s, 1H), 7.37 – 7.32 (m, 2H), 7.19 – 7.08 (m, 2H), 6.79 (td, J = 7.0, 5.9, 3.2 Hz, 2H), 6.54 – 6.40 (m, 1H), 3.99 (s, 3H), 3.51 (d, J = 9.6 Hz, 1H), 2.68 (d, J = 9.6 Hz, 1H), 2.40 (d, J = 4.0 Hz, 1H), 2.28 (d, J = 4.1 Hz, 1H), 2.21 (d, J = 10.6 Hz, 1H), 1.52 (ddt, J = 12.2, 8.7, 4.3 Hz, 1H), 1.35 (m, 11H), 1.21 (p, J = 8.0, 6.4 Hz, 1H), 0.76 (s, 9H), 0.58 (td, J = 10.4, 9.8, 5.5 Hz, 1H).

¹³C NMR (151 MHz, Chloroform-*d*) δ 167.7, 166.9, 163.5, 146.1, 143.5, 141.5, 141.4, 138.1, 137.1, 135.7, 134.9, 133.7, 132.9, 132.8, 131.6, 130.1, 129.1, 128.9, 128.8, 128.7, 128.3, 127.1, 126.8, 126.6, 126.1, 125.9, 125.9, 125.5, 125.4, 123.2, 122.7, 122.2, 121.2, 52.7, 52.3, 49.4, 45.3, 41.6, 37.6, 35.6, 34.6, 31.4, 30.9, 30.5, 30.4.

HRMS (ESI-TOF) calcd for C₄₉H₅₁N₂O₄⁺ ([M+H]⁺): 731.3843, found: 731.3843.

¹H NMR (600 MHz, Chloroform-*d*)

¹³C NMR (151 MHz, Chloroform-d)

HRMS (ESI-TOF)

Sample Name
User Name
Sample Type
ACQ Method

Sample45
Sample
MSMS240326.m

Position P1-E1
Inj Vol 0.2
IRM Calibration Status Some Ions Missed
Comment

Instrument Name Instrument 1
InjPosition
Data Filename JL-C48-4-1-MSMS.d
Acquired Time 3/26/2024 4:15:02 PM (UTC+08:00)

REVIEWERS' COMMENTS

Reviewer #2 (Remarks to the Author):

The authors have addressed the main points raised by the reviewers to a good degree. I therefore recommend to accept this paper without further revisions.

Reviewer #3 (Remarks to the Author):

In the revised manuscript, Zhang, Shi and colleagues provided detailed explanations, additional experiments, recent references, and so on to reply the reviewers' questions. Therefore, I strongly support the acceptance of this manuscript in Nature Communications.